# Accurate Determination of Conversion Gains of SVOM VT CCDs Based on a Signal-Dependent Charge-Sharing Mechanism

**Yue Pan [1,2], Xuewu Fan [1,*], Hu Wang [1], Hui Zhao [1], Yulei Qiu [3], Wei Gao [1] and Jian Zhang [1]**

[1] Xi'an Institute of Optics and Precision Mechanics, Chinese Academy of Sciences, Xi'an 710119, China; panyue@opt.cn (Y.P.); wanghu@opt.ac.cn (H.W.); zhaohui@opt.ac.cn (H.Z.); gaowei@opt.ac.cn (W.G.); zhangjian@opt.ac.cn (J.Z.)

[2] University of Chinese Academy of Sciences, Beijing 100049, China

[3] National Astronomical Observatories, Chinese Academy of Sciences, Beijing 100012, China; qiuyl@nao.cas.cn

[*] Correspondence: fanxuewu@opt.ac.cn

**Abstract:** The signal-variance method and the photon transfer curve method are the most valuable tools for calculating the conversion gains of charge-coupled device (CCD) detectors. This paper describes the phenomena that arise in the conversion gain measurements of space multi-band variable object monitor (SVOM) visible telescope (VT) CCDs, where the results of the signal-variance method increase with the image gray level, and the results of the photon transfer curve method appear with nonlinearity, which is caused by the signal-dependent charge sharing mechanism of back-illuminated CCDs. A numerical simulation model based on random variables was adopted to analyze the influence of the mechanism on the gain determination. The model simulates all the signals and noise in the flat field image, including the photon signal and photon-shot noise, readout noise, fixed pattern noise, and the signal-dependent charge-sharing signal, and it demonstrated agreement with the experimental data. Then, we proposed a quadratic polynomial curve-fitting formula for the photon transfer curve, and we quantitatively analyzed the relationship between the fitting coefficients and the gain, the signal-dependent charge sharing coefficient, and the full well capacity using the control variable method. Finally, the formula was used to accurately determine the conversion gains of SVOM VT CCDs.

**Keywords:** photon transfer curve; charge coupled device (CCD); CCD characterization; conversion gain; SVOM VT

## 1. Introduction

Space multi-band variable object monitor (SVOM) [1,2] is a proposed Chinese–French astronomical satellite, dedicated to the detection, localization, and measurement of gamma-ray bursts (GRBs), while visible telescope (VT) [3,4] is a visible and near-infrared instrument onboard the SVOM. There are two simultaneous channels in VT, which are the 400–650 nm, named the Blue Channel, and 650–1000 nm, named the Red Channel [5,6]. The detector for the blue channel is an Advanced Inverted Mode Operation (AIMO), back-illuminated, basic processed E2V charge-coupled device (CCD)42-80 device with a mid-band antireflection coating, and, for the red channel, it is a Non-Inverted Mode Operation (NIMO), back-illuminated, basic processed E2V CCD42-80 device with an extended red coating manufactured on deep depleted silicon.

The conversion gain of CCD [7,8] is a parameter that characterizes the relationship between the number of photoelectrons generated by CCD and the gray level of the image with the unit of e/DN. Conversion gain is the basis of many CCD photoelectric parameters, such as quantum efficiency, readout noise, dark current, full well capacity, and so on [9]. Thus, it is one of the most important parameters for VT high accuracy photometry [10].

Based on the Poisson statistical distribution of the signal, the most common tools to measure the conversion gain are the signal-variance method and the photon transfer curve method.

However, when using the signal-variance method in the VT gain measurements, the gain test results are inconsistent with the gray level of the image, as shown in Table 1. When using the photon transfer curve method [11], an obvious nonlinear phenomenon appeared with the increase in the gray level, as shown in Figure 1. There is a serious deviation between the two methods. Previous studies by Downing [12,13] proposed that this phenomenon is caused by the signal-dependent charge sharing mechanism of back-illuminated CCDs even though the CCDs have excellent signal linearity.

**Table 1.** Gain measurement results of visible telescope (VT) charge-coupled devices (CCDs) with the signal variance method.

| | Advanced Inverted Mode Operation (AIMO) CCD for VT Blue Channel | | Non-Inverted Mode Operation (NIMO) CCD for VT Red Channel | |
|---|---|---|---|---|
| | Gray Level (DN) | Gain (e/DN) | Gray Level (DN) | Gain (e/DN) |
| 1 | 10,339 | 1.4426 | 10,792 | 1.6541 |
| 2 | 16,599 | 1.4476 | 16,311 | 1.6756 |
| 3 | 21,591 | 1.4556 | 20,611 | 1.6986 |
| 4 | 27,854 | 1.4653 | 25,349 | 1.7203 |
| 5 | 30,651 | 1.4699 | 30,019 | 1.7383 |
| 6 | 37,082 | 1.4720 | 38,359 | 1.7680 |
| 7 | 40,050 | 1.4791 | 40,361 | 1.7821 |
| 8 | 46,111 | 1.4946 | 45,356 | 1.8037 |
| 9 | 49,005 | 1.5011 | 49,884 | 1.8230 |

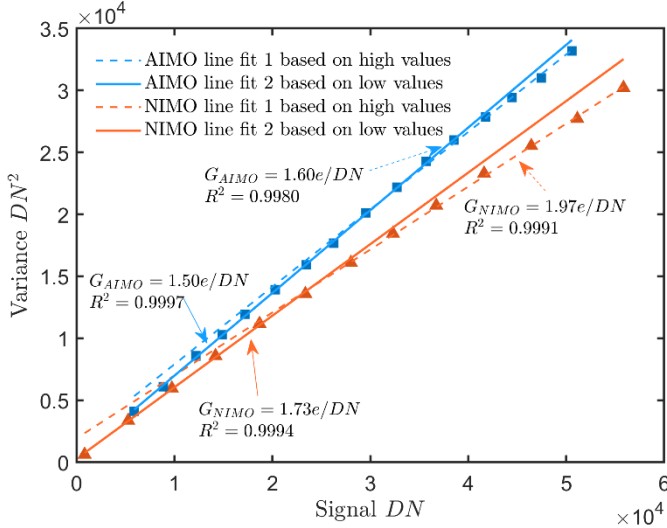

**Figure 1.** Gain measurement results for VT CCDs with the photon transfer curve method.

To accurately determine the conversion gains of SVOM VT CCDs, we present a brief explanation of the signal-dependent charge sharing mechanism in Section 2; in Section 3, a numerical simulation model based on random variables is adopted to analyze the mechanism, and a quadratic polynomial curve-fitting formula is proposed for the photon transfer curve. Finally, in Section 4, based on the quantitative relationship between the fitting coefficients and the gain as well as the other related parameters, the formula is adopted to the determination of conversion gains of SVOM VT CCDs. A summary is provided in Section 5.

## 2. Signal-Dependent Charge Sharing Mechanism

The schematic diagram of the signal-dependent charge-sharing mechanism is shown in Figure 2, which contains three adjacent pixels, each of which includes a charge generation



region, charge transfer region, and potential well region. The charge generation and collection process of a CCD is divided into three stages. First, the incident photons generate electrons on the backside of CCD by the photoelectric effect. Then, the electrons are transferred to the potential well driven by the electric field. Finally, the electrons are collected by the potential well, as shown in Figure 2a.

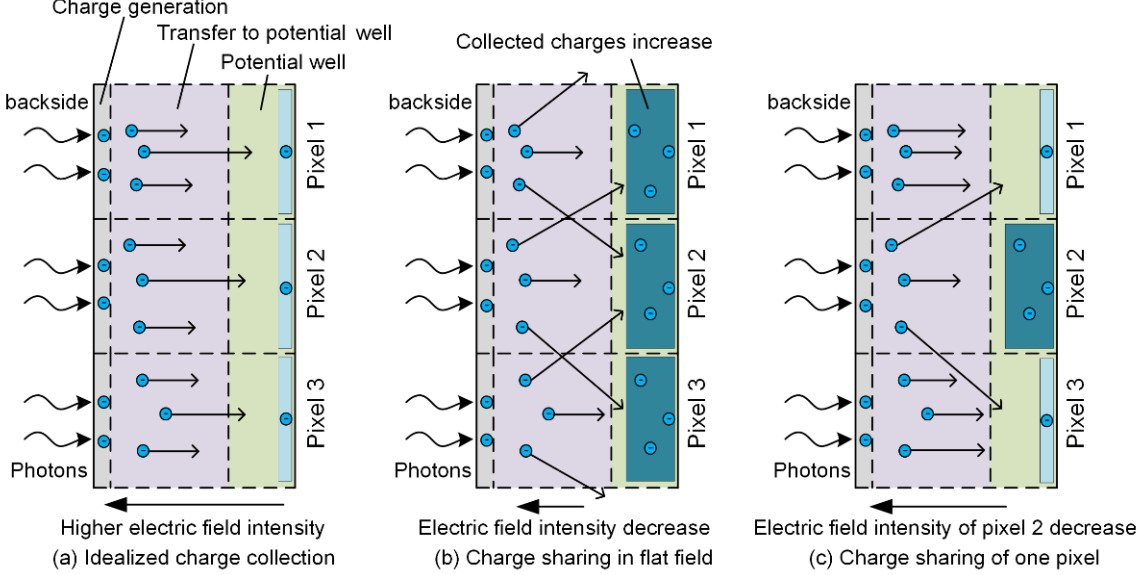

**Figure 2.** Schematic diagram of the signal-dependent charge-sharing mechanism in a back-illuminated CCD.

With the increase in the number of electrons collected in the potential well, the intensity of the driving electric field will gradually weaken, and the binding force on electrons will also decrease. Therefore, during the transfer process, electrons will have a certain probability to transfer to adjacent pixels, resulting in the decrease in image variance, as shown in Figure 2b. The charge sharing is not random but rather has a trend. Pixel that lacks charge will most likely be the receiver of the shared charge, whereas the one having the charge excess will be the donator, as shown in Figure 2c. Thus, the more electrons that are collected, the more serious the charge-sharing phenomenon will be. Different from the decrease in image variance caused by the overflow of electrons when the collected electrons reach the full well, the charge-sharing phenomenon occurs before the electrons enter the potential well and occurs far before saturation.

Under ideal conditions without charge sharing, as shown in Figure 2a, the electrons generated by each pixel are collected by the potential well of that pixel. Since the incident photons in the flat field conform to the Poisson distribution, the number of electrons collected by each potential well also conforms to the Poisson distribution. Therefore, the variance of the number of electrons is equal to the average value of the number of electrons, which is the basis for the signal variance method to calculate the gain. However, as the number of charges collected in the potential well increases, the charge-sharing phenomenon occurs. As shown in Figure 2b, the variance of the number of electrons will be less than the average value, which leads to a higher gain measurement result of the signal variance method and nonlinearity of the photon transfer curve.

Therefore, when using the signal-variance method to calculate the gain, the effect of charge sharing should be reduced. For example, the images with a low gray level [13] can be adopted for gain calculation because the charge-sharing phenomenon is not serious in those images. However, at the same time, the influence of the readout noise should be avoided. Therefore, the number of electrons corresponding to the average gray level of the image should be much larger than the readout noise. Another option is the pixel binning method [12] where the number of electrons collected by each pixel will not be too much, avoiding the charge-sharing phenomenon effectively. In this way, the total number

of electrons collected by the binned pixels is relatively large, which is much larger than the readout noise.

## 3. Numerical Simulation Model

However, whether using low gray-level images or the pixel binning method, there is no quantitative analysis of the influence of charge sharing on the gain calculation results. Therefore, further research is needed.

Konstantin [14] proposed a Monte Carlo model based on the signal-dependent charge sharing mechanism during charge collection to explain the nonlinearity of the photon transfer curve. However, the simulation method is too complex, and there is no further analysis of the effect of the charge sharing coefficient on the gain determination. For the flat field image, we can skip the intermediate process of each photoelectron moving to the potential well affected by the charge-sharing mechanism and consider the result state instead. In the flat field image, we assume the charge-sharing ratio of each pixel to the periphery as $p$, where $p$ is related to charge-sharing coefficient $\beta$, full well capacity $FW$, and the collected electrons $n$, as shown below:

$$p = \frac{n}{FW}\beta. \tag{1}$$

Then, a numerical simulation model based on random variables was adopted to analyze the signal and noise of each pixel. The specific process is shown as follows:

(1) Generate the random variable of electron signal as $p_{e,i}$, where $p_{e,i}$ obeys the Poisson distribution, and the average and variance of $p_{e,i}$ are both equal to the collected electrons $n$:

$$\begin{cases} \overline{p_{e,i}} = n \\ \sigma^2(p_{e,i}) = n \end{cases}. \tag{2}$$

(2) Generate the random variable of readout noise signal as $ron_{e,i}$, while $ron_{e,i}$ obeys the normal distribution with the expectation of 0, and the standard deviation of $RON$ is

$$\begin{cases} \overline{ron_{e,i}} = 0 \\ \sigma(ron_{e,i}) = RON \end{cases}. \tag{3}$$

(3) Generate a random variable of the fixed pattern noise signal as $pn_{e,i}$, while $pn_{e,i}$ obeys the normal distribution with the expectation of 0 and the standard deviation of $P_N$:

$$\begin{cases} \overline{pn_{e,i}} = 0 \\ \sigma(pn_{e,i}) = P_N \end{cases}. \tag{4}$$

(4) The random variable of the total signal can be expressed as:

$$\begin{cases} s_{0,e,i} = (1 + pn_{e,i}) \cdot p_{e,i} + ron_{e,i} \\ s_{0,DN,i} = \frac{(1 + pn_{e,i}) \cdot p_{e,i} + ron_{e,i}}{G} \end{cases} \tag{5}$$

where $s_{0,e,i}$ and $s_{0,DN,i}$ are the total signals in units of electrons and the gray level, respectively.

(5) Considering the effect of charge sharing, each pixel shares the charge of proportion $p$ to the adjacent pixels and receives charges from the adjacent pixels, and the total signal can be expressed as:

$$\begin{cases} s_{e,i} = s_{0,e,i}(1 - p_i) + s_{0,e,i+1}\frac{p_{i+1}}{2} + s_{0,e,i-1}\frac{p_{i-1}}{2} \\ s_{DN,i} = \frac{s_{e,i}}{G} \end{cases}. \tag{6}$$

(6) To obtain the photon transfer curve, the average signal and the signal variance should be calculated. Two flat field images are simulated first as $s_{1,DN,i}$ and $s_{2,DN,i}$ by Equation (6). Then, the average signal $S$ and signal variance $\sigma^2_{shot}$ can be obtained by:

$$
\begin{cases}
S = \frac{\overline{s_{1,DN,i}} + \overline{s_{2,DN,i}}}{2} \\
\sigma^2_{shot} = \frac{\sigma^2(s_{1,DN,i} - s_{2,DN,i})}{2} - \left(\frac{RON}{G}\right)^2
\end{cases}.
\tag{7}
$$

(7) Finally, we set the simulation parameters as the number of pixels $i_{max}$ = 100,000, the gain $G$ = 2e/DN, the readout noise $RON$ = 5e, the fixed pattern noise quality factor $P_N$ = 2%, and the full well capacity $FW$ = 100 ke. The collected electrons $n$ range from 1 to 100 ke in an incremental step of 6 ke, and the charge-sharing coefficient $\beta$ = [0, 0.04, 0.1, 0.2].

The generated photon transfer curve plots for the numerical simulation with four different values of the charge sharing coefficient $\beta$ are shown in Figure 3. The plots for $\beta > 0$ are clearly nonlinear, and we found a quadratic polynomial curve-fitting formula was an excellent fit to the simulated data as:

$$
\sigma^2_{shot} = \gamma S + \nu S^2
\tag{8}
$$

where $\gamma$ is the first-order fitting coefficient, representing the reciprocal of the gain, and $\nu$ is the second-order fitting coefficient, representing the nonlinear parameter of the photon transfer curve.

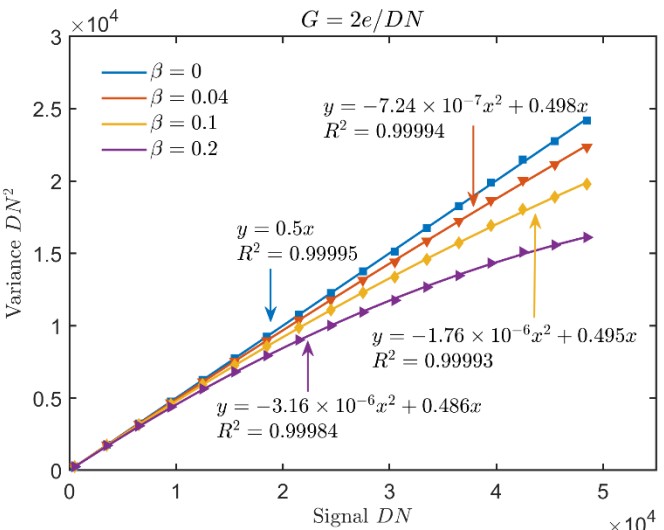

**Figure 3.** Numerical simulation model based on random variables of the photon transfer curve.

However, as can be seen in Figure 3, with the change of charge-sharing coefficient $\beta$, the first-order fitting coefficient $\gamma$ will also change. Taking the reciprocal of $\gamma$ as the gain measurement result, there is still a certain systematic error with the real value. By further analyzing the relationship between the charge-sharing coefficient $\beta$, the first-order coefficient $\gamma$, and the second-order coefficient $\nu$, the gain test result can be corrected.

Next, the relationship between the fitting coefficients and the gain, the coefficient of signal-dependent charge sharing $\beta$, and the full well capacity was quantitative analyzed using the control variable method, as shown in Figure 4. The results show that the first-order fitting coefficient $\gamma$ was related to both the charge-sharing coefficient $\beta$ and the conversion gain but not to the full well capacity, while the second-order fitting coefficient $\nu$ is both the charge-sharing coefficient $\beta$ and the full well capacity but not the conversion gain.

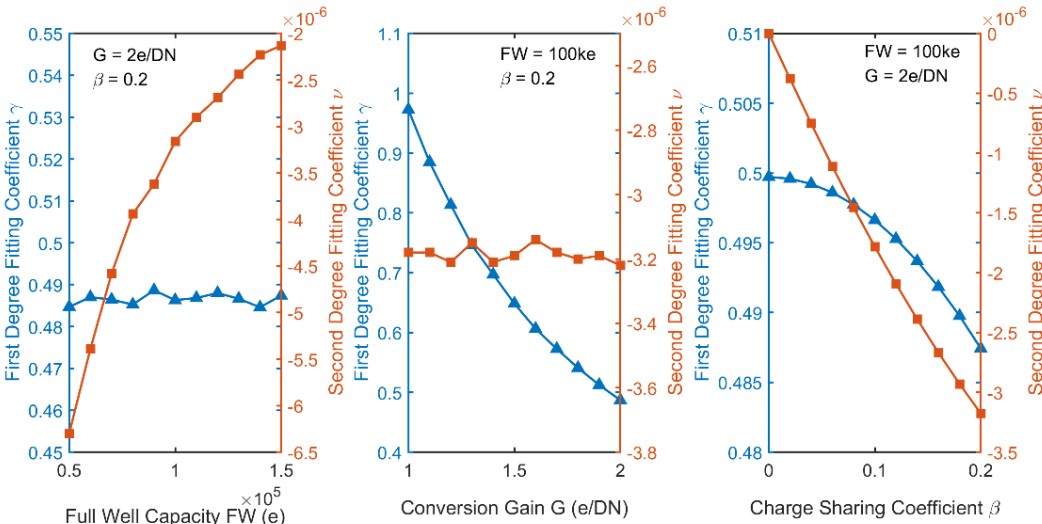

**Figure 4.** The fitting coefficients as functions of full well capacity, conversion gain and charge sharing coefficient.

Next, the curve fitting method was used to determine the numerical relationships.

(1) For the first-order fitting coefficient $\gamma$, the numerical relationship with charge-sharing coefficient $\beta$ and the conversion gain $G$ can be expressed as:

$$\gamma = f_{\gamma,G}(G)f_{\gamma,\beta}(\beta). \tag{9}$$

We considered that in the flat field image, the simulation results of electron loss or electron reception for each pixel should be completely consistent; therefore, the fitting curve of the first-order fitting coefficient $\gamma$ and charge-sharing coefficient $\beta$ should be symmetrical about the $y$-axis. When $\beta = 0$, $\gamma = \frac{1}{G}$. Thus, the quadratic curve is used to fit the relationship between $\gamma$ and $\beta$:

$$\begin{cases} \gamma = \frac{1}{G} \cdot \left(a_\gamma \cdot \beta^2 + 1\right) \\ a_\gamma = -0.6374 \\ R^2 = 0.9974 \end{cases}. \tag{10}$$

(2) For the second-order fitting coefficient $\nu$, the numerical relationship with the charge-sharing coefficient $\beta$ and the full well capacity can be expressed as:

$$\nu = f_{\nu,FW}(FW)f_{\nu,\beta}(\beta). \tag{11}$$

Considering that the fitting coefficient $\nu$ should be 0 when the charge diffusivity $\beta$ is 0, the quadratic curve is used to fit the relationship between them, which is expressed as:

$$\begin{cases} \nu = \frac{a_v}{FW} \cdot \left(b_v\beta^2 + \beta\right) \\ a_v = -2.021 \approx -2 \\ b_v = -0.965 \approx -1 \\ R^2 = 0.9999 \end{cases}. \tag{12}$$

According to Equations (8), (10) and (12), the quadratic curve-fitting formula of the photon transfer curve can be finally expressed as:

$$\sigma^2_{shot} = \frac{\left(1 - 0.6374\beta^2\right)}{G}S - \frac{2\left(\beta - \beta^2\right)}{FW}S^2. \tag{13}$$

The measurement result of the conversion gain with charge-sharing correction is

$$G = \frac{1}{\gamma}\left(1 - 0.6374\beta^2\right) \tag{14}$$

while $\beta$ can be determined using Equation (15)

$$\beta = \frac{1 - \sqrt{1 + 2\nu FW}}{2}. \tag{15}$$

## 4. Experiment

The conversion gain measurements for the detectors were carried out on the SVOM VT CCD test bench [6] in the laboratory shown in Figures 5 and 6. The test bench was composed of an integrating sphere, a Xenon lamp for monochromatic light, a plasma lamp for stable light, a variable attenuator, a monochromator, two picoammeters, two photodiodes for irradiance detection, and a dark box. The special vacuum refrigeration tank could provide temperature changes from −170 degrees to +100 degrees, and the vacuum degree was less than $1 \times 10^{-3}$ Pa, which fully met the working requirements of the CCD detectors. The main characteristics of the CCDs, including bias field images, dark field images, flat field images, conversion gain, readout noise, quantum efficiency, linearity, and so on, were tested to provide an objective assessment of the performance of the CCD detectors.

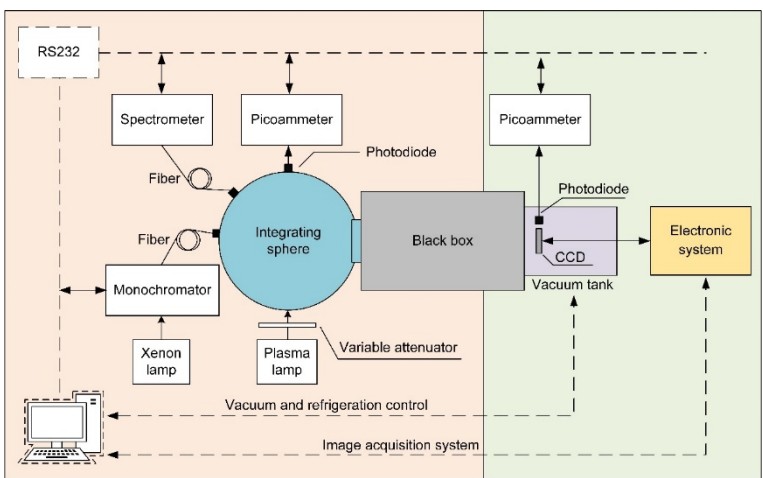

**Figure 5.** Structure diagram of the space multi-band variable object monitor (SVOM) VT CCD test bench.

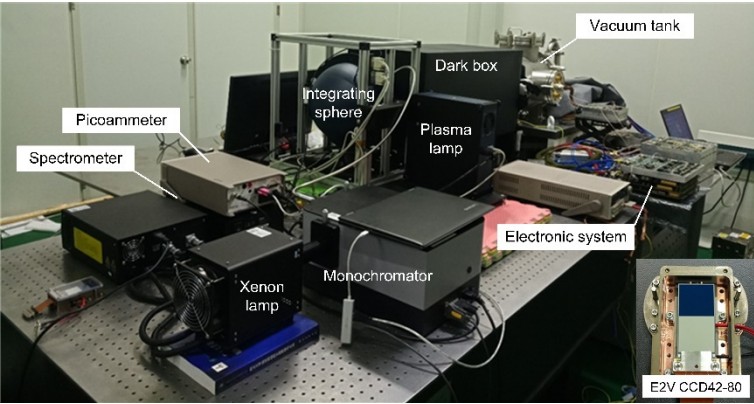

**Figure 6.** Overview of the SVOM VT CCD test bench.

In the process of the gain measurements, two bias images were acquired first with the light source off. Then, the plasma lamp was turned on and worked in stable mode with the stability better than 99.8%. A set of flat field images in different gray levels were acquired by adjusting the variable attenuator, among which two images were acquired for each gray level.

The average value of the signal under each gray level can be calculated by subtracting the sum of the average values of the two bias images from the sum of the average values of the two flat field images and then dividing by 2, as shown in Equation (16).

$$\overline{S} = \frac{(\overline{flat_1} + \overline{flat_2}) - (\overline{bias_1} + \overline{bias_2})}{2} \tag{16}$$

Upon subtracting one flat field image from the other one, the new image eliminated the fixed pattern noise and contained shot noise and readout noise [15]. Subtracting one bias image from the other one, the new image contained only readout noise. Then, the signal variance under each gray level can be calculated by subtracting the variance of the former image from the latter one, and then dividing by 2, as shown in Equation (17).

$$\sigma^2 = \frac{\sigma_{flat_1 - flat_2}{}^2 - \sigma_{bias_1 - bias_2}{}^2}{2} \tag{17}$$

The signal variance as a function of the average signal within the linear photo response range of the CCD is shown in Figure 7 for VT CCDs. The dependences are clearly nonlinear, which is similar to the simulated results in Figure 3. A fit to the quadratic polynomial curve-fitting Equation (8) was applied to the experimental data and achieved excellent agreement.

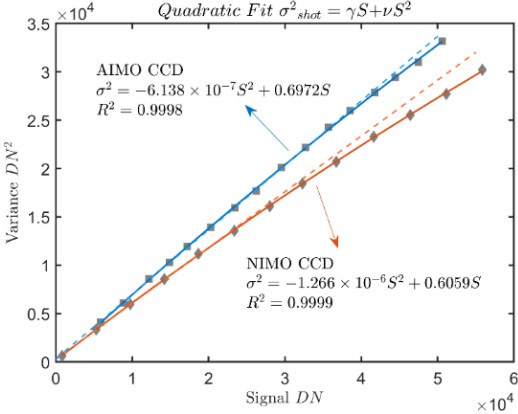

**Figure 7.** The photon transfer curves of VT CCDs, fitted with the quadratic function.

The full well capacities of the AIMO CCD and NIMO CCD were about 100 ke and 120 ke in the datasheet, respectively, and thereby, the conversion gain was calculated from the relationship between the fitting coefficients and the gain, the signal-dependent charge sharing coefficient, and the full well capacity simulated in Equations (14) and (15). The results are as follows.

For the AIMO CCD of VT blue channel:

$$\begin{cases} \gamma = 0.6972 \\ \nu = -6.138 \times 10^{-7} \\ \beta = 0.0317 \\ G_{AIMO} = 1.4334 \, e/DN \end{cases} \tag{18}$$

For the NIMO CCD of VT red channel:

$$\begin{cases} \gamma = 0.6059 \\ \nu = -1.266 \times 10^{-6} \\ \beta = 0.0828 \\ G_{NIMO} = 1.6432 \, e/DN \end{cases} \tag{19}$$

Thus, from the above calculation results, the conversion gains of SVOM VT CCDs were 1.4334 e/DN for the blue channel AIMO CCD and 1.6432 e/DN for the red channel NIMO CCD, respectively.

According to the test results, the nonlinearity of the photon transfer curve was related not only to the full well capacity but also to the charge-sharing coefficient $\beta$. The charge-sharing coefficient $\beta$ depends on the device parameters, such as the thickness and the collection phase voltage as well as the resistivity of the silicon. The full well capacities of the AIMO CCD and NIMO CCD were 100 and 120 ke, respectively, and the difference between the reciprocal of the FWs was about 17%. The two CCDs used the same test circuit and the same silicon material, and thus, the biggest difference was the thickness of silicon. The AIMO CCD was a 16 μm standard silicon, while the NIMO CCD was a 40 μm deep depletion type. There was approximately a 150% difference. Therefore, the nonlinearity of the photon transfer curve tended to increase with the thickness of the photosensitive silicon.

## 5. Summary

To accurately determine the conversion gains of SVOM VT CCDs, this paper presents a brief explanation of the signal-dependent charge sharing mechanism of back-illuminated high precision scientific CCDs. A numerical simulation model based on random variables was adopted to analyze the mechanism, and a quadratic polynomial curve-fitting formula was proposed for the photon transfer curve. Finally, the formula was applied to the experimental data and achieved excellent agreement. Based on the quantitative relationship between the fitting coefficients and the gain as well as the other related parameters simulated in this paper, the conversion gains of SVOM VT CCDs were determined. The derived quadratic fit to the photon transfer curve can be used for more robust calculation of the conversion gain in scientific CCDs.

**Author Contributions:** Conceptualization, Y.P.; methodology, Y.P.; software, H.Z.; validation, X.F.; formal analysis, H.W.; investigation, Y.Q.; resources, J.Z.; data curation, W.G.; writing—original draft preparation, Y.P.; writing—review and editing, Y.P. All authors have read and agreed to the published version of the manuscript.

**Funding:** This research was funded by the National Natural Science Foundation of China [Grant No. 61107008 & Grant No. 61105017].

**Acknowledgments:** The authors thank the outstanding work of the SVOM VT team from Xi'an Institute of Optics and Precision Mechanics of CAS, National Astronomical Observatories of CAS, and Innovation academy for microsatellites of CAS.

**Conflicts of Interest:** The authors declare no conflict of interest.

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
