# Peer review of "Accurate Determination of Conversion Gains of SVOM VT CCDs Based on a Signal-Dependent Charge-Sharing Mechanism"

_electronics, doi:10.3390/electronics10080931_

Round 1
Reviewer 1 Report
general remarks
-have a native speaker check the sentences
-nice short paper and to the point
-if possible, try to prevent abbreviations
Some detailed remarks.
Add correlation coefficient to fig 1 and fig 3 and fig 7
31-32 Just a comment: 400-650nm is almost white since this spectrum contains blue+green+red.
118 equation 2 average of a poison is λ not λ2
138 formula 7: the simulation you did must have RON=0 otherwise a different fit-formula is needed on your simulated image.
Equation 15 is correct in that it compensates for the readnoise and only shotnoise is measured.
152-153 figure 4: In formula 1 the ratio β/FW determines p. There is no formula in your paper with only β or FW. So, one expects dependence on β/FW only. Why does first degree fitting coefficient not depend on FW (fig 4a) but depend on β (fig 4c)?
Note: A quick check of equation 2 and 6 for small beta tells me that the fit should read: σ2shot = S/G - β /FW*S. According the paper (equation 7, 9, 11 ) it is about σ2shot = S/G - 4* β /FW*S. Can you explain?
Reviewer 2 Report
- Within the abstract in lines 12-14 need rephrasing
- The overall model has to be revised, since it does not consider the charge a certain pixel receives from its neighbors. In other words, a pixel can "share" 3 electrons to its surroundings, but receive 2 or 3 or even 4 back from other pixels. To remain within the same model, one can consider gradual lightning of the focal array, where he/she can assume a certain direction/gradient/probability of sharing. In case the illumination is uniform, the model for 2/3/4 pixels has to be developed.
- Consequently, the measurements have to be carried once again to verify the modified model
Reviewer 3 Report
It would be useful to have an additional section before the section 5 summarizing the physical information derivable from the experimental resultrs obtained.
Reviewer 4 Report
The authors have proposed a quadratic model for conversion gains of CCDs based on the charge sharing mechanism. This paper provides some well-presented discussions and experimental data, however, I have some questions about the charge sharing mechanism. I think it is worthy of publication once the authors answer the following questions.
- The charge sharing mechanism is caused by a decrease in the electric field. Can you discuss more about the electric field decrease mechanism? It looks like an electric field collapse from the space charge effect.
- If the electric field decreases because of the space charge effect, the electric field distribution will collapse in the middle of the transfer region. The electric field distribution is not flat, and the decrease in the electric field is not proportional to the collected electron n. Therefore, for equation (1), the sharing ratio p is not a linear function of n.
- In figure 3, how did you simulate the data, is it based on equation (6)? If so, would you mind telling us the values you used in these equations, such as λ, RON, and PN?
- The equation (6) is a 1D pixel model, which only considers the left and right adjacent pixels. For real applications, it should be a 2D pixels. It will introduce top, bottom, and diagonal adjacent pixels. Please comment why equation (6) is a good assumption.
- The NIMO CCD has a bigger second order shot noise coefficient than the AIMO CCD. Could you explain why? Actually, the NIMO CCD even has a bigger FW 120ke, second order coefficient should be lower as shown in equation (11).
Round 2
Reviewer 2 Report
For the better understanding of the charge sharing effect, it is worthwhile explaining that the charge share is not random, but rather has a trend, e.g. pixel that lacks charge will most likely be the receiver of the charge, whereas the one having the charge excess will be the donator. This way it is much easier to perceive that the average of the given pixel will remain the same, while the variance will drop. Adding a picture will also help.
2. line 140- it should read "receives..."
Reviewer 4 Report
The authors have added more detail to the paper to make their conclusions more supportable. I think it is worth publishing now.
Author Response
Thank you very much for your comments and recognition!